# Combination of Irinotecan and Melatonin with the Natural Compounds Wogonin and Celastrol for Colon Cancer Treatment

**DOI:** 10.3390/ijms24119544

**Published:** 2023-05-31

**Authors:** Anna Radajewska, Helena Moreira, Dorota Bęben, Oliwia Siwiela, Anna Szyjka, Katarzyna Gębczak, Paulina Nowak, Jakub Frąszczak, Fathi Emhemmed, Christian D. Muller, Ewa Barg

**Affiliations:** 1Department of Basic Medical Sciences, Wroclaw Medical University, 50-556 Wroclaw, Poland; anna.szyjka@umw.edu.pl (A.S.); katarzyna.gebczak@umw.edu.pl (K.G.); ewa.barg@umw.edu.pl (E.B.); 2Department of Medical Laboratory Diagnostics, Division of Clinical Chemistry and Laboratory Hematology, Wroclaw Medical University, 50-556 Wroclaw, Poland; 3The Hubert Curien pluridisciplinary Institute, UMR 7178 Centre National de la Recherche Scientifique (CNRS), University of Strasbourg, 67081 Illkirch, France; fathi.emhemmed@iphc.cnrs.fr (F.E.); cdmuller@unistra.fr (C.D.M.); 4Faculty of Pharmacy, Wroclaw Medical University, 50-556 Wroclaw, Poland; dorota.biegas@student.umw.edu.pl (D.B.); oliwia.siwiela@student.umw.edu.pl (O.S.); paulina.nowak@student.umw.edu.pl (P.N.); jakub.fraszczak@student.umw.edu.pl (J.F.)

**Keywords:** colon cancer, irinotecan, melatonin, celastrol, wogonin

## Abstract

Colorectal cancers are one of the leading cancers worldwide and are known for their high potential for metastasis and resistance to therapy. The aim of this study was to investigate the effect of various combination therapies of irinotecan with melatonin, wogonin, and celastrol on drug-sensitive colon cancer cells (LOVO cell line) and doxorubicin-resistant colon cancer stem-like cells (LOVO/DX cell subline). Melatonin is a hormone synthesized in the pineal gland and is responsible for circadian rhythm. Wogonin and celastrol are natural compounds previously used in traditional Chinese medicine. Selected substances have immunomodulatory properties and anti-cancer potential. First, MTT and flow cytometric annexin-V apoptosis assays were performed to determine the cytotoxic effect and the induction of apoptosis. Then, the potential to inhibit cell migration was evaluated using a scratch test, and spheroid growth was measured. The results showed important cytotoxic effects of the drug combinations on both LOVO and LOVO/DX cells. All tested substances caused an increase in the percentage of apoptotic cells in the LOVO cell line and necrotic cells in the LOVO/DX cell subline. The strongest effect on the induction of cancer cell death was observed for the combination of irinotecan with celastrol (1.25 µM) or wogonin (50 µM) and for the combination of melatonin (2000 µM) with celastrol (1.25 µM) or wogonin (50 µM). Statistically significant improvements in the effect of combined therapy were found for the irinotecan (20 µM) and celastrol (1.25 µM) combination and irinotecan (20 µM) with wogonin (25 µM) in LOVO/DX cells. Minor additive effects of combined therapy were observed in LOVO cells. Inhibition of cell migration was seen in LOVO cells for all tested compounds, while only irinotecan (20 µM) and celastrol (1.25 µM) were able to inhibit LOVO/DX cell migration. Compared with single-drug therapy, a statistically significant inhibitory effect on cell migration was found for combinations of melatonin (2000 µM) with wogonin (25 µM) in LOVO/DX cells and irinotecan (5 µM) or melatonin (2000 µM) with wogonin (25 µM) in LOVO cells. Our research shows that adding melatonin, wogonin, or celastrol to standard irinotecan therapy may potentiate the anti-cancer effects of irinotecan alone in colon cancer treatment. Celastrol seems to have the greatest supporting therapy effect, especially for the treatment of aggressive types of colon cancer, by targeting cancer stem-like cells.

## 1. Introduction

Colorectal cancers (CRCs), i.e., cancers of the colon and rectum, are among the world’s leading cancers. CRCs are the third most commonly diagnosed neoplasm in men, particularly in developed countries. Despite the availability of well-established surgical treatment, adjuvant chemotherapy, and the possibility of early detection, the mortality rate is still very high [1,2,3]. In addition, there is still the problem of the toxicity of treatment and the chemoresistance for which cancer stem cells (CSCs) may be responsible [4]. One of the recently used drugs for solid tumors is irinotecan. The metabolite of irinotecan is an inhibitor of topoisomerase-I, which is involved in DNA replication. However, it has been discovered that the rate of metabolism and pharmacokinetics of this drug differ between individuals. Irinotecan is commonly used with other drugs, such as 5-fluorouracil, oxaliplatin, or monoclonal antibodies [5]. However, treatment with irinotecan and its combination with 5-fluorouracil and leucovorin can cause serious neutropenia or diarrhea that affects the comfort of the patient’s life and may require the discontinuation of therapy [6]. Natural compounds such as green tea extract have been proven to reduce the risk of carcinogenesis [7]. Studies on better delivery methods of drugs and natural compounds are also ongoing. Nanoparticles allow for the delivery of more stable therapeutics to increase the cytotoxic effect of a drug or natural compound [8,9]. Plant-derived substances have been used in traditional Chinese and Indian medicine for centuries [10]. They can bring positive effects to cancer therapy by reducing the toxicity of chemotherapy and via the synergic or additive effect of the supplemented compound [10]. Therefore, the combination of other components with irinotecan could have a potentially positive effect. 

Melatonin is a hormone produced mainly in the pineal gland but also in other parts of the human body, including the gastrointestinal tract [11,12]. It regulates circadian rhythm and has been proven to have an immunomodulatory effect, anti-oxidant properties, and anti-cancer properties [13,14]. Melatonin has a protective effect on healthy cells during cancer treatment [11]. Studies have also confirmed the synergic effect of melatonin and anti-cancer drugs [14,15]. 

Wogonin is a natural flavonoid derived from plants, mainly the roots of *Scutellaria baicalensis Georgi*. The dried roots have been used in China for inflammatory diseases, hepatitis, and bacterial and viral infections [16,17]. Currently, studies on wogonin are focused on its anti-cancer properties, ability to activate apoptosis, inhibitory effect on proliferation, invasion, and angiogenesis, as well as its induction of reactive oxygen species (ROS) production and autophagy [18].

Celastrol is another promising substance that has been used in traditional Chinese medicine for centuries. Celastrol, also known as Tripterine, is a terpenoid derived from the roots of the *Tripterygium wilfordii Hook F.* plant (popularly called Thunder God Vine). It demonstrates anti-inflammatory, anti-diabetic, and neuroprotective capacities. In addition, it has attracted attention in the context of cancer treatment [19]. Celastrol influences the angiogenesis and metastasis of cancer and suppresses tumor growth [20]. Celastrol has been proven to inhibit colon cancer by regulating TGF-β1/Smad signaling [21], reducing NO production and thereby controlling angiogenesis [20], and causing cell cycle arrest with the generation of double-stranded breaks [22]. 

In this study, we aimed to evaluate the anti-cancer effect of a combination of natural plant-derived substances (celastrol and wogonin) with standard anti-cancer drugs—irinotecan and melatonin—on metastatic colon cancer (LOVO cell line) and cancer stem-like cells (LOVO/DX cell line) (Figure 1). 

## 2. Results

### 2.1. Effect of Tested Compounds and Their Combinations on the Viability of Colon Cancer Cells and Cancer Stem-Like Cells

Drug-sensitive (LOVO) and drug-resistant stem-like (LOVO/DX) colon cancer cells were treated individually with irinotecan (5 µM or 20 µM), melatonin (2000 µM), celastrol (0.625 µM or 1.25 µM), or wogonin (25 µM or 50 µM), or with a selected combination of these compounds, for 72 h. Cell viability was assessed using the MTT assay. The results are presented in Figure 2, Figure 3 and Figure 4. As shown in Figure 2, all tested compounds have potent cytotoxic properties against both LOVO (panel A) and LOVO/DX cells (panel B). The effect of melatonin is comparable to that of irinotecan (at a concentration of 20 µM); both compounds reduce cell viability by about 45–50%. Celastrol causes a 60–62% (LOVO) and 72–75% (LOVO/DX) decrease in cell viability, while wogonin decreases it by 50–70% (LOVO) and 68–80% (LOVO/DX).

In LOVO cells (Figure 3), the combination of melatonin with irinotecan (20 µM), celastrol (1.25 µM), or wogonin (50 µM) significantly improved the cytotoxic effects of melatonin alone. Irinotecan at 5 µM, celastrol at 0.625 µM, and wogonin at 25 µM all potentiated the effects of melatonin: however, the results were not statistically significant. Furthermore, melatonin (2000 µM) used in combination with irinotecan (5 or 20 µM) inhibited cell viability more strongly compared with treatment with irinotecan alone. The combination of wogonin or celastrol with irinotecan (20 µM) also improved the effect of irinotecan. 

In LOVO/DX cells (Figure 4), a statistically significant increase in cytotoxicity compared with melatonin was found for combinations of melatonin with celastrol (1.25 µM) or wogonin (50 µM). Other combinations caused a small increase in the frequency of dead cells compared with melatonin alone. Compared with treatment with irinotecan alone, wogonin (25 and 50 µM) and celastrol (0.625 and 1.25 µM) improved the effect of 20 µM irinotecan, and melatonin improved the effect of 5 µM irinotecan. 

In both cell lines, the highest level of dead cells was noted for the combination of irinotecan with celastrol or wogonin and melatonin with celastrol (Table 1).

### 2.2. Pro-Apoptotic and Necrotic Activity of Tested Compounds and Their Combinations in Colon Cancer Cells and Cancer Stem-Like Cells

The ability of the tested compounds and their combinations to induce apoptosis in LOVO and LOVO/DX cells was assessed after 72 h of treatment using annexin-V and propidium iodide staining (PI). The frequency of apoptotic cells (annexin-V+/PI_−_ and annexin-V+/PI+) and necrotic cells (annexin-V_−_/PI+) was measured by flow cytometry (Appendix A). 

First, we evaluated the effects of individual compounds, i.e., irinotecan, melatonin, celastrol, and wogonin, on the frequency of apoptotic and necrotic cells (Figure 5). All compounds showed comparable pro-apoptotic effects in both cell lines. The effect on drug-sensitive cells (LOVO) (Figure 5A) was much stronger than on drug-resistant cancer stem-like cells (LOVO/DX) (Figure 5B). In contrast, only LOVO/DX was sensitive to the necrotic effects of the compounds. However, it should be emphasized that the compounds have a low necrotic potential (about 10% of necrotic cells), except irinotecan, which induces necrosis in approximately 24% of cells. The effects were, however, statistically significant for all compounds.

In LOVO cells (Figure 6), the combination of melatonin with irinotecan, celastrol, or wogonin slightly improved the proapoptotic effects of melatonin alone. The level of apoptotic cells increased by 10, 12, 9, 2, and 13% for irinotecan (5 and 20 µM), celastrol (0.625 and 1.25 µM), or wogonin (25 µM), respectively. Compared with irinotecan alone, the proapoptotic effects were comparable. None of the tested compounds or their combinations had an important impact on the necrosis of LOVO cells.

In LOVO/DX cells (Figure 7), celastrol (1.25 µM) and wogonin (25 µM) combined with melatonin improved the effect of melatonin by 14 and 7%, respectively. Importantly, a statistically significant increase in the frequency of apoptotic cells was noted for the combination of irinotecan (20 µM) with celastrol (1.25 µM) or wogonin (25 µM), compared with irinotecan alone. Additionally, a reduction in the level of necrotic cells was noted for celastrol, melatonin, and wogonin used in combination with irinotecan. There was no impact of the tested drug combinations on cell necrosis compared with melatonin, except for irinotecan and celastrol, which slightly increased the percentage of necrotic cells compared with melatonin used alone. The percentage of apoptotic cells treated with the combination of tested drugs is summarized in Table 2.

### 2.3. Inhibition of Cell Migration by Tested Compounds and Their Combinations in Colon Cancer Cells and Cancer Stem-Like Cells

Cell migration is one of the important factors cancer metastasis and increases the risk of patient death. The potential of the selected drugs and their combinations to inhibit tumor metastasis was tested using a wound-healing or scratch assay. Combined therapy and single-drug treatment were applied to LOVO and LOVO/DX cells for 48 h. Table 3 presents results for LOVO and LOVO/DX cells treated with combinations of the selected compounds. Figure 8, Figure 9, Figure 10, Figure 11 and Figure 12 show pictures of scratches and the results.

As demonstrated in Figure 8, with the exception of 5 µM irinotecan and 2000 µM melatonin, treatment with single drugs significantly inhibited the migration of LOVO cells. Migration was completely blocked by 20 µM irinotecan, as detected by an increase in the scratch width compared with the control (shown as a negative percentage on the graph). The combination of irinotecan (5 µM) with wogonin (25 µM) showed a significantly greater inhibitory effect on LOVO cell migration compared with either irinotecan or wogonin alone. Celastrol in combination with irinotecan 20 µM improved the cell migration inhibitory effect of celastrol alone but not of irinotecan alone. A significantly stronger inhibitory effect was detected for a combination of melatonin (2000 µM) with wogonin (25 µM) compared with melatonin alone, however, when compared with wogonin alone, the percentage of scratch closure was significantly higher for this combination. 

LOVO/DX cells showed a much lower migratory rate compared with LOVO cells, with a gap closure of about 30% (Figure 9). Irinotecan (20 µM) and celastrol (1.25 µM) significantly decreased the capacity of LOVO/DX cells to migrate. A statistically significant inhibitory effect was detected for the combination of melatonin (2000 µM) and wogonin (25 µM) compared with melatonin or wogonin alone. In addition, a strong inhibitory effect was detected after using irinotecan (5 or 20 µM) in combination therapy with wogonin (25 µM) and celastrol (0.625 µM and 1.25 µM) compared with wogonin or celastrol, respectively. However, none of the compounds improved the effect of irinotecan.

### 2.4. Effect of Combined Therapy on Spheroid Growth

We performed the spheroid creation assay on a ultra-low attachment plate (ULA plate) and observed the spheroid growth for two days. Pictures were taken every 6 h during the experiments. We evaluated the changes in spheroid size after 48 h and compared them with sizes at time 0, but no significant changes were found. We observed that LOVO cells formed more dense cultures with a spherical structure, while LOVO/DX cells created less compact aggregates. The results are shown in Figure 13, Figure 14 and Figure 15.

## 3. Discussion

Irinotecan alone or in combination with 5-fluorouracil (5-FU) and folic acid (FA) is used for the treatment of metastatic colorectal cancer [27]. It has several side effects, such as diarrhea, neutropenia, alopecia, nausea, vomiting, and acute cholinergic-like syndrome. The active metabolite of irinotecan, SN-38, is metabolized and inactivated by the liver enzyme UDP-glucuronosyltransferase 1A1 (UGT1A1). Even a mild deficiency of this enzyme, as seen in congenital Gilbert’s syndrome, leads to toxic effects of irinotecan therapy [28]. Therefore, new treatment strategies to overcome these limitations are still being sought. Among the investigated strategies are (1) antibody–drug conjugates and small-molecule drug conjugates; (2) gene therapy, such as epigenetic approaches, noncoding RNA, CRISPR/Cas9, and antisense oligonucleotides; (3) protein inhibitors; (4) utilization of well-known drugs, e.g., anti-inflammatory drugs such as metformin; and (5) combination therapy with herbal substances [29,30]. In addition, to overcome the common problem with drug delivery and bioavailability, other drug delivery systems are being evaluated, such as hydrogels, liposomes, exosomes, and nanoparticles [29]. One interesting strategy is gut microbiota modulation via probiotics, prebiotics, antibiotics, or fecal microbiota transplantation, which showed promising effects but has to be further investigated [31]. Here, we evaluated whether the use of irinotecan with natural compounds (melatonin, celastrol, and wogonin) may improve the treatment of metastatic colon cancer and eliminate cancer stem-like cells. Studies were carried out using two colon cancer cell lines: (1) the LOVO cell line, a drug-sensitive colon cancer cell line derived from a metastatic site (the left supraclavicular region) of a colorectal cancer patient, and (2) the LOVO/DX cell subline, consisting of doxorubicin-resistant colon cancer stem-like cells enriched in CSCs that have been used as an in vitro cellular model of aggressive and resistant colon cancer [32,33].

Melatonin is a naturally occurring substance in the human body and a very important hormone for homeostasis, including the proper functioning of the colon and peristaltic movements. Melatonin has been proposed to play a key role in preventing carcinogenesis in the intestine. It is not clear whether plasma or urine melatonin levels correlate with cancer risk [34]. However, multiple pre-clinical studies confirmed the anti-cancer and anti-inflammatory properties of melatonin. Zhang et al. demonstrated that melatonin consumption correlates with a decreased risk of CRC cancer in older patients [35]. In our studies, we found that melatonin used in monotherapy was able to eliminate both drug-sensitive and CSC-like cells. In addition, in LOVO cells, melatonin induced apoptosis, but not necrosis, and inhibited cell migration. Both weak pro-apoptotic and necrotic effects were observed in LOVO/DX cells, but no effect was observed on cell migration. 

In addition, we found a positive effect of combined therapy with melatonin and irinotecan. We observed an increased number of dead cells in the LOVO line when treated with melatonin and irinotecan (5 µM and 20 µM). However, the effectiveness of this combination was not observed for cell migration and induction of apoptosis. The combination of melatonin with irinotecan did not inhibit migration or significantly improve the cytotoxic and pro-apoptotic effects of the treatment in LOVO/DX cells. 

Melatonin was described as having anti-oxidative properties [36]. Different anti-oxidants, such as vitamins A, E, C, and melatonin, were tested for their ability to improve irinotecan therapy in colorectal adenocarcinoma cells and non-small cell lung cancer [37]. Vitamins A and E improved irinotecan activity in apoptosis induction, while vitamin C did not show any positive effect on irinotecan therapy. Similar to our findings, melatonin did not enhance the pro-apoptotic effect of irinotecan. However, the selected concentration of melatonin in the study by Kontek et al. was much lower (50 µM) than in our study (2000 µM) [37]. Melatonin was previously studied as a supportive cancer therapy. Melatonin was proven to enhance ionizing radiation (IR) treatment both in vitro and in vivo [38]. Treatment with melatonin and IR suppressed colony formation and migration in HCT 116 cells and increased apoptosis and DNA damage. Combined therapy led to cell cycle arrest in the G2/M phase, which is known as the phase most sensitive to radiation. Furthermore, the tumor growth rate, as well as tumor volume, was reduced for melatonin and IR groups compared with melatonin or ionizing radiation treatment alone [38]. 

Chinese medicine has been widely investigated and used to support patient well-being during anti-cancer therapy. For instance, the herbal mixture ‘Huang-Qin’—a preparation from the root of *Scutellaria baicalensis Georgi* containing, among others, wogonin, baicalin, wogonoside, and baicalein—has been used to reduce vomiting, diarrhea, and nausea. In addition, the modified formulation of Huang-Qin-PHY906 has been proven to enhance the effects of several anti-cancer drugs [39]. In our study, we showed that wogonin has a strong anti-cancer effect, mainly on LOVO cells. Treatment with wogonin alone increased the percentage of apoptotic cells compared with the control, without changing the percentage of necrotic cells. In addition, wogonin significantly inhibited LOVO cell migration. In contrast, in LOVO/DX cells, wogonin slightly increased the percentage of apoptotic and necrotic cells but had no impact on cell migration. However, our previous study showed that wogonin decreased the percentage of the side population (SP), which is enriched in cancer stem cells, in both LOVO and LOVO/DX cells [40]. Wang et al. noted that hydrophobic flavonoids such as baicalein and wogonin decrease the proliferation of colon cancer cells in a dose-dependent manner, induce cell cycle arrest in the S and G2/M phases, and promote cell death via apoptosis. They proposed that apoptosis is partly induced through the mitochondrial pathway, since the mitochondrial membrane potential drops after treatment of colon cancer cells with baicalein and wogonin [41]. The combined treatment of established cancer therapy and wogonin has been analyzed for various solid tumors such as hepatocellular carcinoma [42], ovarian cancer [43], and head and neck tumors [44]. Wogonin with sorafenib has a higher cytotoxic potential than single compound therapy alone. Wogonin sensitizes hepatocellular carcinoma cells to sorafenib, leading to increased cytotoxicity and apoptosis activation. The autophagy that supports the cytoprotective action of sorafenib treatment was reduced by combined therapy with wogonin [42]. 

We showed that the use of wogonin with irinotecan increased the anti-cancer activity of irinotecan in both LOVO and LOVO/DX cells. Wogonin improved the cytotoxic effects of irinotecan therapy in LOVO cells more strongly than in LOVO/DX cells. We also observed an increase in the percentage of apoptotic cells as well as decreased cancer cell migration after treatment with both compounds,. A stronger pro-apoptotic effect was observed for the LOVO cell line; however, the combined treatment did not reach a significant difference when treated with a single drug or two drugs together. LOVO/DX treated with irinotecan (20 µM) with wogonin (25 µM) resulted in a higher percentage of apoptotic cells compared with a single irinotecan treatment. According to our study, a combination of a low dose of irinotecan (5 µM) with wogonin significantly reduced LOVO cell migration. Cell migration was also inhibited in LOVO/DX cells when irinotecan and wogonin were used in combination, though less effectively. Melatonin treatment was also improved by the addition of a high dose of wogonin (50 µM), as demonstrated by the decrease in cancer cell viability and a slight increase in the percentage of apoptotic cells. Moreover, the combination of melatonin and wogonin (25 µM) significantly influenced LOVO and LOVO/DX cell migration compared with melatonin alone. Wogonin has been proven to inhibit cancer cell migration and invasion, which suggests that, besides its anti-cancer properties, it has a strong anti-metastatic feature [17,45]. It was proven that wogonin has a positive effect on rat thymocytes during etoposide treatment [46]. The adverse effect of etoposide was manifested with the induction of apoptosis in healthy cells such as rat thymocytes and mouse bone marrow cells. The increased nuclear condensation, DNA fragmentation, and caspase activation induced by etoposide were abolished by wogonin. In addition, wogonin improved the anti-cancer activity of etoposide for Jurkat cells and A549 cells but did not change the etoposide effect on HL-60 and NCI-H226 cells [46]. 

Celastrol is a substance that has been used for centuries in traditional Chinese medicine as an anti-inflammatory and anti-cancer drug. The high incidence of cancer and the current problem with drug resistance development prompted us to look for combined therapies with a cancer cell-sensitization effect. Celastrol has been proven to induce apoptosis in several tumors, and it prevents invasion and metastasis at a low IC50 concentration. In our study, treatment with celastrol causes LOVO and LOVO/DX apoptosis and increases the percentage of necrotic LOVO/DX cells. As we have previously shown, the pro-apoptotic effect of celastrol is partly related to the high concentration of ROS in the cytosol and mitochondria [33]. In this study, we used a slightly lower concentration of celastrol alone or in combination with irinotecan or melatonin and a longer incubation with cells (72 or 48 h). We demonstrated that the combination of irinotecan and celastrol increased the number of dead cells in both the LOVO and LOVO/DX cell lines. In addition, the combination of 1.25 µM celastrol and 20 µM irinotecan increased the rate of apoptotic cells in the LOVO/DX cell line. LOVO/DX cells are thought to be enriched with CSCs because they express multiple proteins and biomarkers commonly assigned to cancer stem cells. Therefore, it better represents aggressive colon cancer. However, we did not observe an effect of the combination of celastrol with irinotecan on cancer cell migration, which was mainly due to the very strong effect of irinotecan (20 µM). Celastrol has a strong effect on LOVO/DX cells when used alone. The combination of melatonin and celastrol (1.25 µM) increased the percentage of dead cells for the LOVO line. LOVO/DX cells treated with the same combination responded with a higher percentage of dead cells when compared with single melatonin use. Melatonin combined with a low dose of celastrol (0.625 µM) also decreased the positive effect of 0.625 µM celastrol treatment for LOVO/DX cells. Similar tendencies were observed for cell migration of LOVO/DX cells where melatonin in combination with celastrol (1.25 µM) reduced the inhibitory effect of celastrol. Zhang et al. showed that celastrol works synergistically with sorafenib (the kinase inhibitor used for the treatment of liver cancer) [47]. It was demonstrated that celastrol with sorafenib increases apoptosis in in vitro and in vivo models and reduces the total growth of transplanted liver tumors [47]. Similarly, in breast cancer, a combination of celastrol with tamoxifen had a synergistic effect on cancer cell development and growth [48]. Enhanced inhibitory effects of treatment with celastrol and tamoxifen on MCF-7 cell proliferation were shown using a colony formation assay; in addition, an increased frequency of apoptosis and activation of autophagy was found. Moreover, MCF-7-implanted tumors in nude mice had smaller volumes when treated with combination therapy compared with single therapy with tamoxifen [48]. 

## 4. Materials and Methods

### 4.1. Reagents

DMEM F12 (Dulbecco’s Modified Eagle’s Medium: Nutrient Mixture F-12, (BE)12-708), HBSS (Hank’s Balanced Salt Solution, BE10-547F), FBS (fetal bovine serum, BI Biological Industries, 04-007-1A), ultraglutamine 1 (Bio Whittaker, BE 17-605E/U1), and gentamicin sulfate were purchased from Lonza (17-518Z, Basel, Switzerland). Accutase™ Cell Detachment solution was obtained from BD Biosciences (BD 561527, Franklin Lakes, NewJersey, USA). Alexa Fluor^®^ 488 Annexin V/Dead Cell Apoptosis Kit was from Thermofisher (V 13245), TrypLE™ Express was from Gibco (cat. 12604054, Waltham, MA, USA), celastrol (>98% purity), was purchased from Cayman Chemical Company (Item No. 70950, AnnArbor, MI, USA). DMSO (D2650), melatonin (M5250), wogonin hydrate (W0769), irinotecan hydrochloride (1347609) and MTT ((3-(4,5-dimethylthiazol-2-yl)-2,5-diphenyltetrazolium bromide, M5655) were obtained from Sigma-Aldrich (St. Louis, MO, USA). Isopropanol (117515002) was from Chempur (Poland).

### 4.2. Cell Culture Conditions

The human colon cancer cell line LOVO was obtained from the American Type Culture Collection (ATCC, Manassas, VA, USA). The doxorubicin-resistant cell line LOVO/DX was derived from the LOVO cells by 3 months of cultivation in the presence of a low concentration of doxorubicin. Both LOVO and LOVO/DX cell lines were cultivated in DMEM F12 medium supplemented with 10% fetal bovine serum (FBS), 2 mM L-glutamine, and 25 μg/mL of gentamicin. Cells were maintained under standard culture conditions (37 °C in a humidified atmosphere with 5% CO_2_) and passaged approximately every 3 days using TrypLE™ Express (GIBCO, Waltham, MA, USA). 

### 4.3. Reagent Preparation

In the present study, the compounds irinotecan, wogonin, celastrol, and melatonin were examined. The following were purchased from Sigma-Aldrich (St. Louis, MO, USA): irinotecan hydrochloride, wogonin hydrate, and melatonin. Celastrol was purchased from Cayman Chemical Company (Ann Arbor, MI, USA). Wogonin, irinotecan, and celastrol were dissolved in DMSO as a 10 mM stock solution and stored at −20 °C. Melatonin was dissolved in DMSO as a 2 M stock solution and stored at −20 °C. Further dilution was performed in cell culture media. Cells were treated with 2 mM of melatonin, 5 or 20 µM of irinotecan, 1.25 or 0.625 µM of celastrol, and 25 or 50 µM of wogonin. The final DMSO concentration in the cell culture did not exceed 0.5% in the highest concentration of the compounds.

### 4.4. Cell Viability Assay

Cell viability was determined using the MTT assay. This assay measures the reduction of a yellow tetrazolium salt (3-(4,5-dimethyl-2-thiazolyl)-2,5-diphenyl-2H-tetrazolium bromide (MTT)) into insoluble formazan by mitochondrial enzymes. Briefly, colon cancer cells (2.5 × 10^4^/200 µL) in each well on the 96-well plates were cultured overnight for better attachment. The next day, the media was replaced with fresh media contained various concentrations of melatonin (2000 µM), irinotecan (5 µM, 20 µM), celastrol (0.625 µM, 1.25 µM), wogonin (25 µM) or a combination of these compounds. Control cells received 0.5% DMSO solution in the culture medium. Cells were incubated for 72 h with drugs or DMSO in conditions of 5% CO_2_, and 95% humidity at 37 °C. After the incubation, the medium was replaced with 1 mg/mL MTT (Sigma-Aldrich, St. Louis, MO, USA) solution and incubated for 2 h. Next, the MTT solution was removed, and isopropanol was added to the cells and incubated in the dark for 30 min. Then, the plate was read at 555 nm by using the Perkin Elmer Wallac 1420 Victor2 microplate reader (GMI, USA). The effect on cell viability was assessed as the percent cell viability compared with DMSO control cells, which were assigned as 100% viability. Drug concentrations and incubation time was established based on a pilot study.

### 4.5. Apoptosis and Necrosis Assay

Apoptosis and necrosis were detected with flow cytometry after staining the cells with a fluorochrome mixture of Annexin V–Alexa Fluor^®^488 and PI, using the Alexa Fluor^®^488 Annexin V/Dead Cell Apoptosis Kit. This staining allows for the discrimination between early and late apoptotic cells and necrotic cells. Cells (1 × 10^6^/mL) were seeded in a 6-well plate and incubated with the various tested compounds (37 °C, 5% CO_2_). Following 72 h of incubation, the cells were detached with Accutase™ Cell Detachment solution and washed with HBSS. The cells were resuspended in 100 μL of ice-cold 1x binding buffer and stained with 5 μL Annexin V–Alexa Fluor^®^488 and 1 μL PI for 15 min in the dark at room temperature. Samples were immediately analyzed with the CyFlow^®^ SPACE flow cytometer (Sysmex, Kobe, Prefektura Hyōgo, Japan). A 488 nm (50 mW) laser excitation and the 536/40 (BP) and 675/20 (BP) filters were used for fluorescence measurement of Alexa Fluor 488 and Propidium iodide, respectively. The results were analyzed using FlowMax (Sysmex, Kobe, Prefektura Hyōgo, Japan).

### 4.6. Spheroid Preparation and Cell Growth Assessment

The effects of the test substances on 3D cell cultures were examined using spheroids [49,50]. Cells from the LOVO and LOVO/DX cell lines were seeded onto a U-bottom plate with a less adherent surface at a rate of 3000 cells per well in a volume of 100 μL DMEM F12 medium. The plate was centrifuged and then incubated for 4 days at 37 °C and 5% CO_2_. After this time, the spheroids were imaged (t = 0 h), and immediately after that, the test substances (twice as concentrated) dissolved in 100 μL of medium were added to 100 μL of the medium in the wells with spheroids, thus obtaining the expected concentrations. Images were then taken every 24 h for 72 h using the automated microscope Incucyte^®^ S3—Automated Live Cell Analysis by Sartorius. 3D culture growth was measured automatically using Incucyte hardware-compatible software.

### 4.7. Scratch Assay

A scratch assay was conducted to evaluate cell migration. A total of 150,000 cells were seeded on the 96-well plates and grown to full confluency [51,52]. Cell monolayers were wounded with the Incucyte^®^ 96-well Woundmaker Tool and washed with PBS. Regular culture medium (DMEM F12) was replaced with a medium containing tested compounds and their combinations and placed in the Incucyte^®^ S3 in the incubator (37 °C, 5% CO_2_). The cell migration was monitored and pictures were taken every 6 h for 3 days (72 h). The mean width of the wound was calculated by the software at time 0 and after 48 h. 100% represents total gap closure. The width of the gap was calculated after 48 h and shown as a percentage of the healed-wound width after 48 h compared with the wound width at time 0. The gap closure was calculated based on the equation (T0 − T48)/T0 ∗ 100%.

### 4.8. Statistical Analysis

All experiments were performed in triplicate. The obtained data were analyzed with GraphPad Prism 8.0.1 for Windows (GraphPad Software, San Diego, CA, USA). Results are shown as the mean value ± standard deviation (SD). To evaluate differences between groups, the Student’s *t*-test was applied. Statistical significance is considered as *p*-values < 0.05.

## 5. Conclusions

Our study showed that all compounds, both in monotherapy and combined, were able to eliminate both drug-sensitive and CSC-like cells. However, they demonstrate different effects on apoptosis, necrosis, and cancer cell migration. Wogonin appeared to be effective alone as well as in combination with irinotecan or melatonin in the suppression of colon cancer cell growth, the induction of apoptosis, and the inhibition of colon cancer cells migration. Therefore, wogonin can be considered a good natural agent for the treatment of colon cancer, while being distinguished by its safety for healthy cells. Celastrol was also proven to have a strong inhibitory effect on colon cancer cell growth. The combination of celastrol with irinotecan seems to be the best treatment for eliminating CSC-like cells, mainly through cytotoxic and pro-apoptotic effects. This combination may be promising for a very aggressive type of colon cancer. Anti-cancer effects of melatonin (cytotoxic and pro-apoptotic) might be improved by celastrol or wogonin in both drug-sensitive and cancer stem-like colon cancer cells. However, a combination of melatonin with celastrol is probably not a good solution for inhibiting tumor invasiveness.

## Figures and Tables

**Figure 1 ijms-24-09544-f001:**
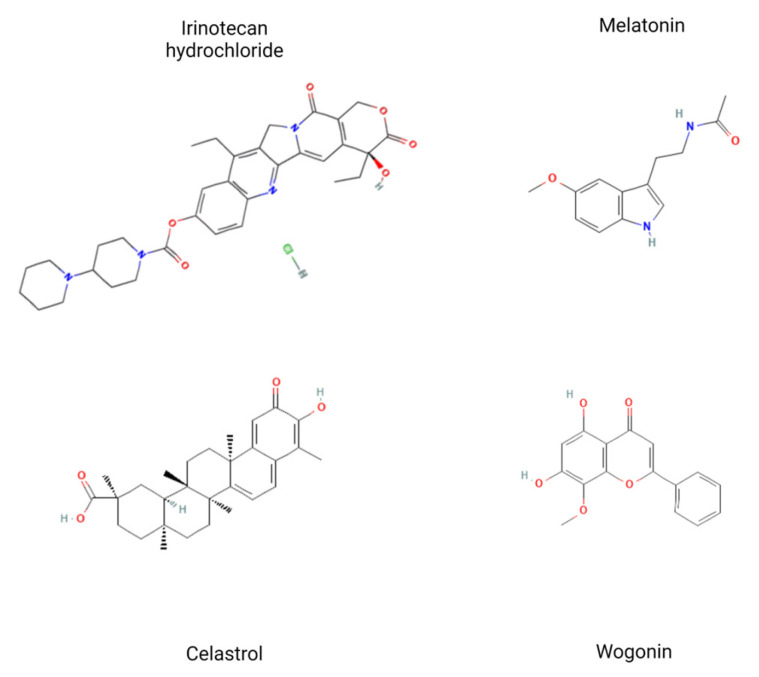
Structural formulas of tested compounds: irinotecan hydrochloride [23], melatonin [24], celastrol [25] and wogonin [26]. Figure prepared with BioRender.

**Figure 2 ijms-24-09544-f002:**
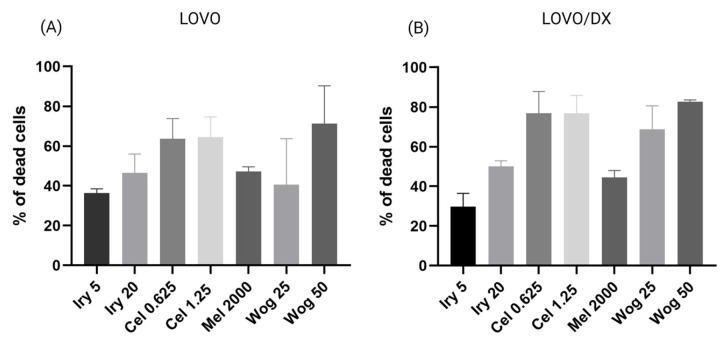
Cytotoxic effects of irinotecan (5 µM and 20 µM), melatonin (2000 µM), celastrol (0.625 µM and 1.25 µM), and wogonin (25 µM and 50 µM) on LOVO (**A**) and LOVO/DX (**B**) cells after 72 h treatment. The results are presented as the mean ± SD of five independent experiments. The significance of the differences was determined by the Mann–Whitney U test. Abbreviations: Iry (irinotecan), Mel (melatonin), Cel (celastrol), Wog (wogonin).

**Figure 3 ijms-24-09544-f003:**
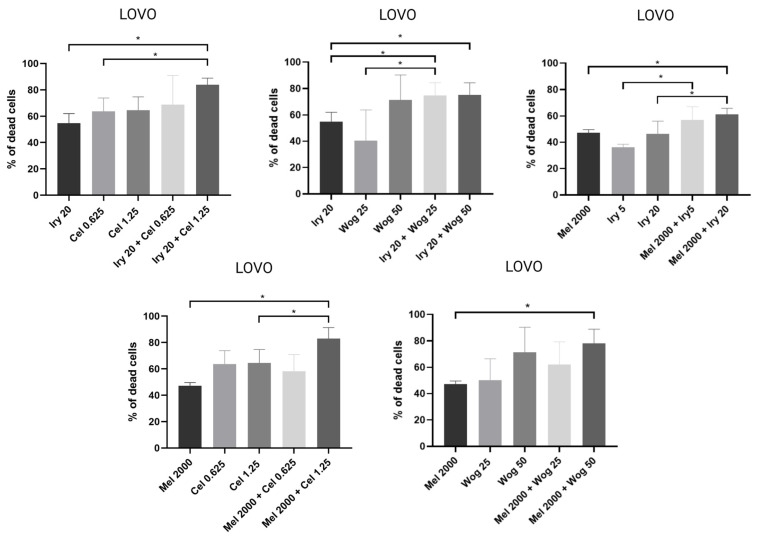
Cytotoxic effects of the combination of irinotecan (5 µM and 20 µM), melatonin (2000 µM), celastrol (0.625 µM and 1.25 µM) and wogonin (25 µM and 50 µM) on LOVO cells after 72 h treatment. The results are shown as the mean ± SD of five independent experiments. The significance of the differences was determined by the Mann–Whitney U test. * *p* < 0.05. Abbreviations: Iry (irinotecan), Mel (melatonin), Cel (celastrol), Wog (wogonin).

**Figure 4 ijms-24-09544-f004:**
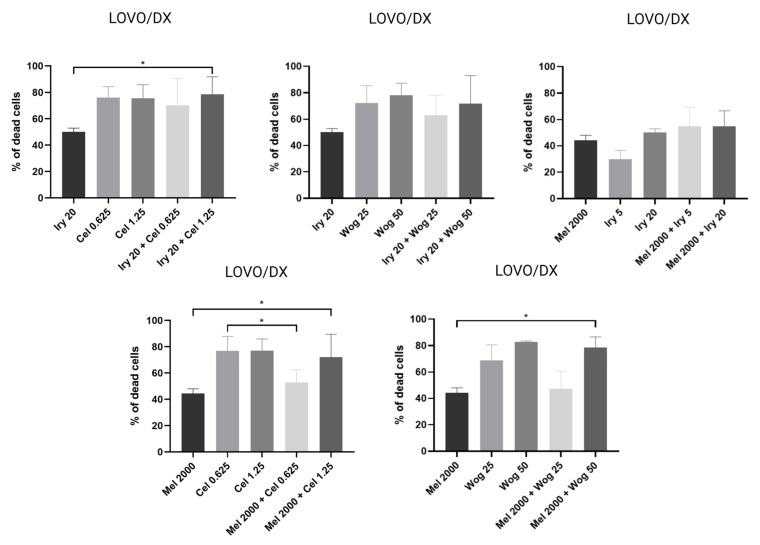
Cytotoxic effects of the combination of irinotecan (5 µM and 20 µM), melatonin (2000 µM), celastrol (0.625 µM and 1.25 µM) and wogonin (25 µM and 50 µM) on LOVO/DX cells after 72 h treatment. The results are shown as the mean ± SD of five independent experiments. The significance of the differences was determined by the Mann–Whitney U test. * *p* < 0.05. Abbreviations: Iry (irinotecan), Mel (melatonin), Cel (celastrol), Wog (wogonin).

**Figure 5 ijms-24-09544-f005:**
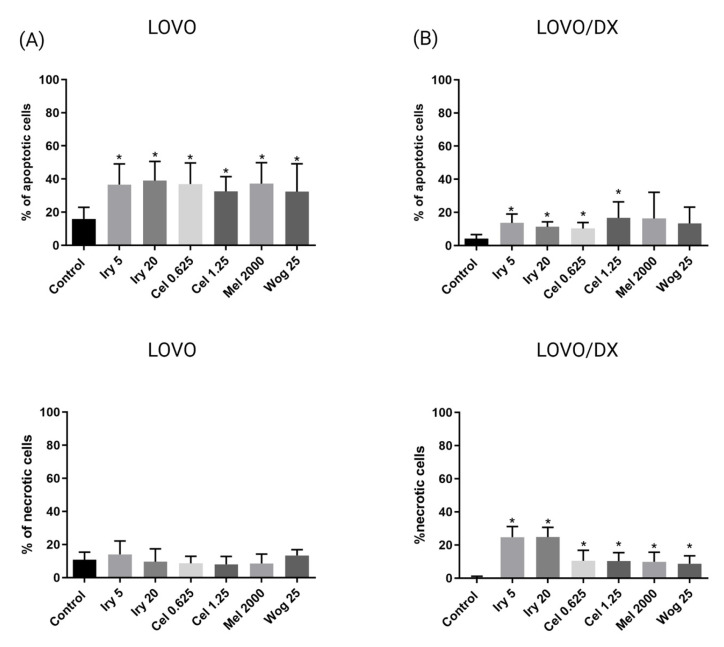
The effects of 72 h of incubation with irinotecan (5 μM and 20 μM), melatonin (2000 µM), wogonin (25 μM), or celastrol (0.625 μM and 1.25 μM) on the frequency of apoptosis and necrosis in LOVO (**A**) or LOVO/DX (**B**) cells. The cells were double-stained with Annexin V–Alexa Fluor^®^488 and PI fluorescent dyes (Alexa Fluor^®^ 488 Annexin V/Dead Cell Apoptosis Kit) and analyzed by flow cytometry. The results are presented as the percentage of apoptotic cells (Annexin V–Alexa Fluor^®^488+ and PI- or Annexin V–Alexa Fluor^®^488+ and PI+) and necrotic cells (Annexin V–Alexa Fluor^®^488- and PI+). The results are presented as the mean ± SD of five independent experiments. The significance of the differences was determined by the Mann–Whitney U test. * *p* < 0.05. Abbreviations: Iry (irinotecan), Mel (melatonin), Cel (celastrol), Wog (wogonin).

**Figure 6 ijms-24-09544-f006:**
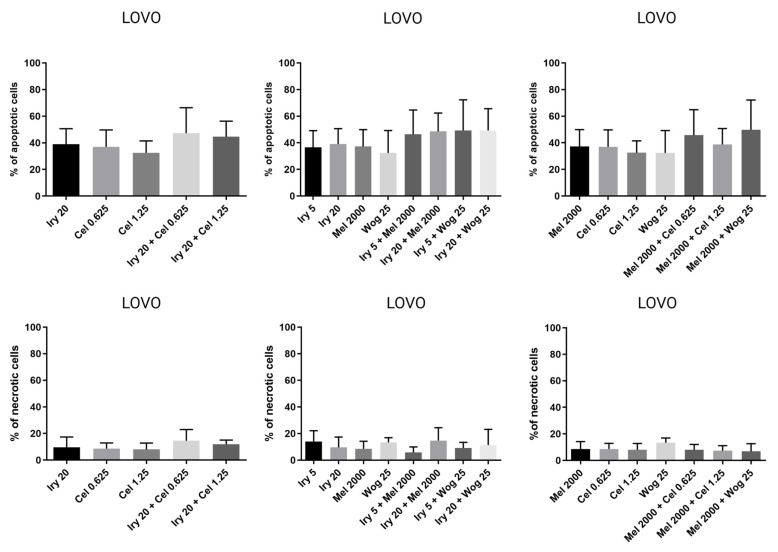
The effects of 72 h of incubation with irinotecan (5 μM and 20 μM), melatonin (2000 µM), wogonin (25 μM), celastrol (0.625 μM and 1.25 μM), or their combinations on the frequency of apoptosis and necrosis in LOVO cells. The cells were double-stained with Annexin V–Alexa Fluor^®^488 and PI fluorescent dyes (Alexa Fluor^®^ 488 Annexin V/Dead Cell Apoptosis Kit) and analyzed by flow cytometry. The results are presented as the percentage of apoptotic cells (Annexin V–Alexa Fluor^®^488+ and PI- or Annexin V–Alexa Fluor^®^488+ and PI+) and necrotic cells (Annexin V–Alexa Fluor^®^488- and PI+). The results are displayed as the mean ± SD of five independent experiments. The significance of the differences was determined by the Mann–Whitney U test. Abbreviations: Mel (melatonin), Iry (irinotecan), Cel (celastrol), Wog (wogonin).

**Figure 7 ijms-24-09544-f007:**
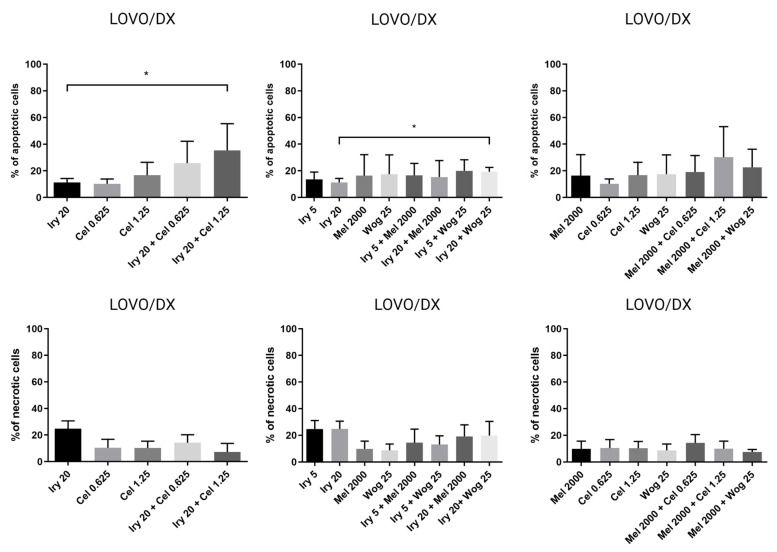
The effect of 72 h of incubation with irinotecan (5 μM and 20 μM), melatonin (2000 µM), wogonin (25 μM), celastrol (0.625 μM and 1.25 μM), or their combinations on the frequency of apoptosis and necrosis in LOVO/DX cells. The cells were double-stained with Annexin V–Alexa Fluor^®^488 and PI fluorescent dyes (Alexa Fluor^®^ 488 Annexin V/Dead Cell Apoptosis Kit) and analyzed by flow cytometry. The results are presented as the percentage of apoptotic cells (Annexin V–Alexa Fluor^®^488+ and PI- or Annexin V–Alexa Fluor^®^488+ and PI+) and necrotic cells (Annexin V–Alexa Fluor^®^488- and PI+). The results are presented as the mean ± SD of five independent experiments. The significance of the differences was determined by the Mann–Whitney U test. * *p* < 0.05. Abbreviations: Mel (melatonin), Iry (irinotecan), Cel (celastrol), Wog (wogonin).

**Figure 8 ijms-24-09544-f008:**
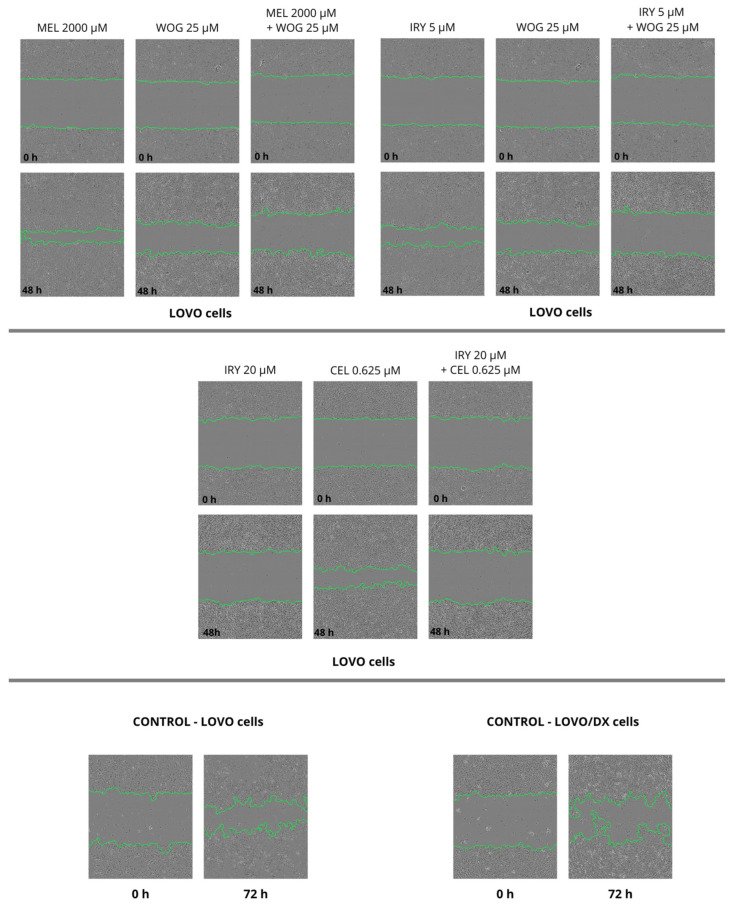
Selected pictures of wound-healing assays for LOVO and LOVO/DX cells after 48 h of incubation with irinotecan (5 μM and 20 μM), melatonin (2000 µM), wogonin (25 μM), celastrol (0.625 μM and 1.25 μM), and their combinations. Abbreviations: Iry (irinotecan), Mel (melatonin), Cel (celastrol), Wog (wogonin).

**Figure 9 ijms-24-09544-f009:**
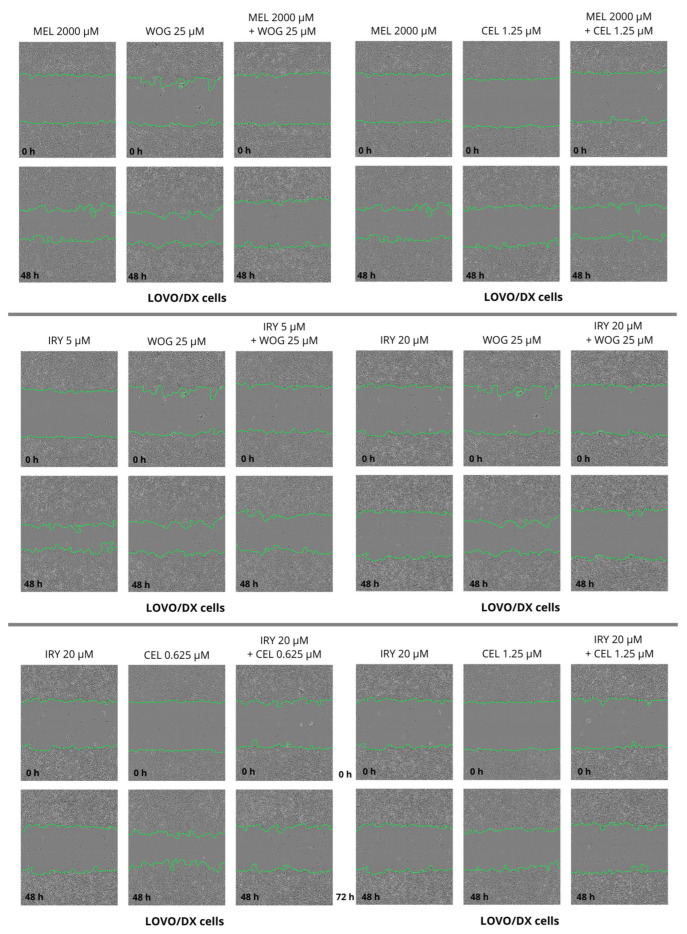
Selected pictures of wound-healing assays for LOVO and LOVO/DX cells after 48 h of incubation with irinotecan (5 μM and 20 μM), melatonin (2000 µM), wogonin (25 μM), celastrol (0.625 μM and 1.25 μM), and their combinations. Abbreviations: Iry (irinotecan), Mel (melatonin), Cel (celastrol), Wog (wogonin).

**Figure 10 ijms-24-09544-f010:**
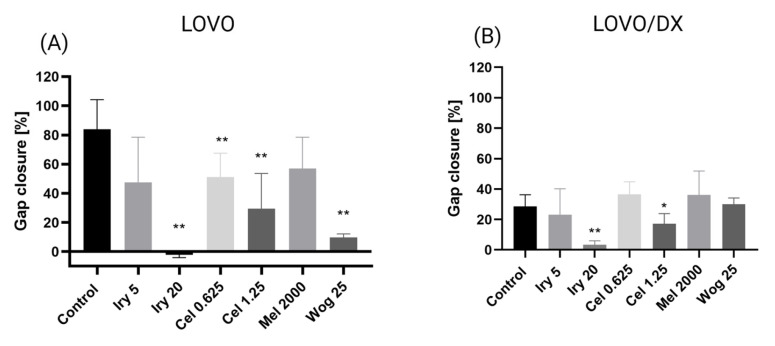
The effect of 48 h of incubation of irinotecan (5 μM and 20 μM), melatonin (2000 µM), wogonin (25 μM) and celastrol (0.625 μM and 1.25 μM) with LOVO (**A**) and LOVO/DX cells (**B**) on the percentage gap closure. The results show the mean ± SD of five independent experiments. The significance of the differences was determined by the Mann–Whitney U test. * *p* < 0.05, ** *p* < 0.01. Abbreviations: Iry (irinotecan), Mel (melatonin), Cel (celastrol), Wog (wogonin).

**Figure 11 ijms-24-09544-f011:**
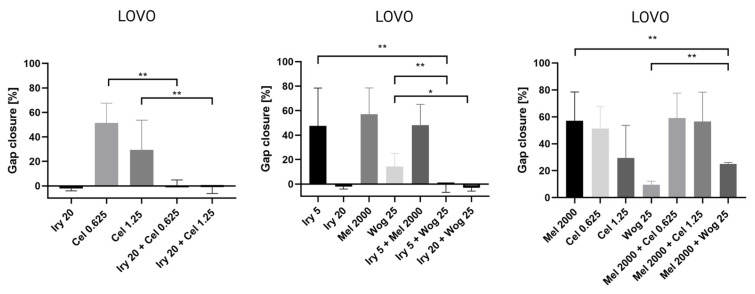
The effect of 48 h of incubation of irinotecan (5 μM and 20 μM), melatonin (2000 µM), wogonin (25 μM), celastrol (0.625 μM and 1.25 μM), and their combinations with LOVO cells on the percentage gap closure. The results are shown as the mean ± SD of five independent experiments. The significance of the differences was determined by the Mann–Whitney U test. * *p* < 0.05, ** *p* < 0.01. Abbreviations: Iry (irinotecan), Mel (melatonin), Cel (celastrol), Wog (wogonin).

**Figure 12 ijms-24-09544-f012:**
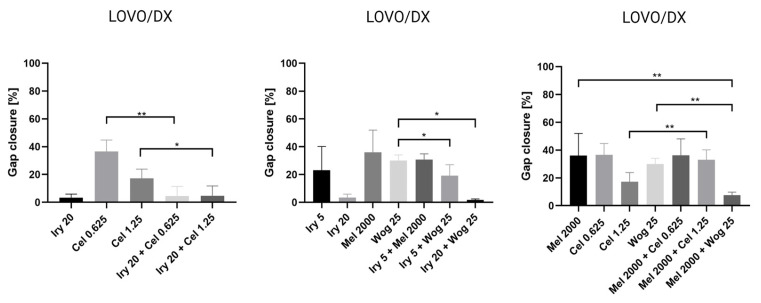
The effect of 48 h of incubation of irinotecan (5 μM and 20 μM), melatonin (2000 µM), wogonin (25 μM), celastrol (0.625 μM and 1.25 μM), and their combinations with LOVO/DX cells on the percentage of gap closure. The results are shown as the mean ± SD of five independent experiments. The significance of the differences was determined by the Mann–Whitney U test. * *p* < 0.05, ** *p* < 0.01. Abbreviations: Iry (irinotecan), Mel (melatonin), Cel (celastrol), Wog (wogonin).

**Figure 13 ijms-24-09544-f013:**
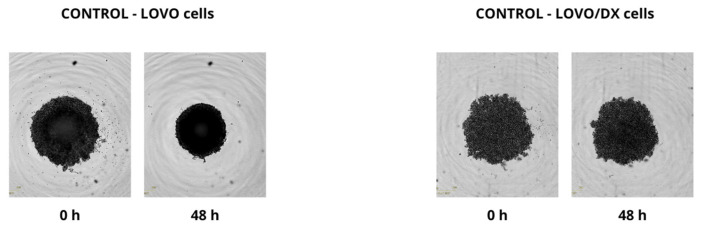
Pictures spheroid growth of the control group (without drug) of LOVO and LOVO/DX cells after 48 h.

**Figure 14 ijms-24-09544-f014:**
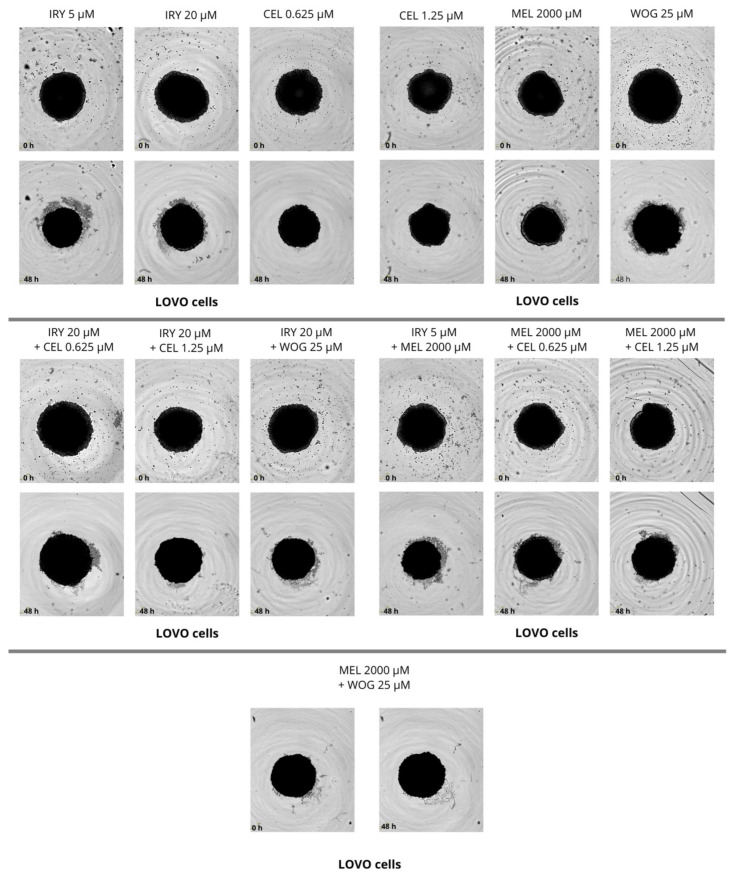
The effect of 48 h of incubation with irinotecan (5 μM and 20 μM), melatonin (2000 µM), wogonin (25 μM), celastrol (0.625 μM and 1.25 μM), and their combinations on the spheroid growth of LOVO cells. Abbreviations: Iry (irinotecan), Mel (melatonin), Cel (celastrol), Wog (wogonin).

**Figure 15 ijms-24-09544-f015:**
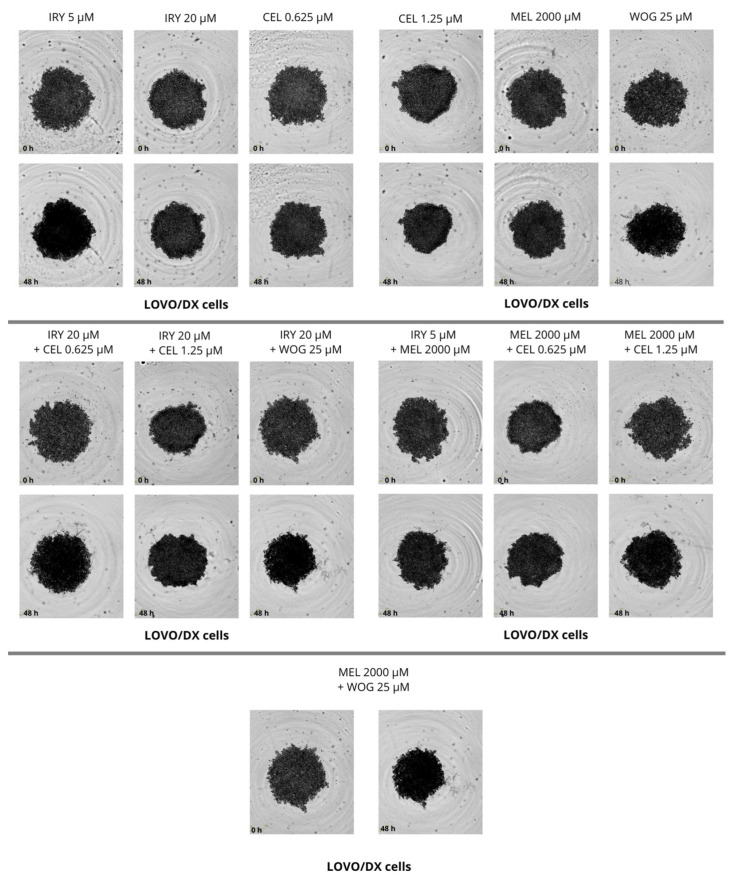
The effect of 48 h of incubation with irinotecan (5 μM and 20 μM), melatonin (2000 µM), wogonin (25 μM), celastrol (0.625 μM and 1.25 μM), and their combinations on the spheroid growth of LOVO/DX cells. Abbreviations: Iry (irinotecan), Mel (melatonin), Cel (celastrol), Wog (wogonin).

**Table 1 ijms-24-09544-t001:** Combination of drugs with the highest level of dead cells. Abbreviations: Iry (irinotecan), Mel (melatonin), Cel (celastrol), Wog (wogonin).

Irinotecan and Melatonin Alone	Percentage of Dead Cells	
	LOVO Cells	LOVO/DX Cells
IRY 5 µM	36%	30%
IRY 20 µM	55%	50%
MEL 2000 µM	47%	44%
Combination of drugs		
IRY 20 µM + CEL 0.625 µM	69%	70%
IRY 20 µM + CEL 1.25 µM	84%	79%
IRY 20 µM + WOG 25 µM	75%	63%
IRY 20 µM + WOG 50 µM	75%	72%
MEL 2000 µM + IRY 5 µM	57%	55%
MEL 2000 µM + IRY 20 µM	61%	55%
MEL 2000 µM + CEL 0.625 µM	58%	53%
MEL 2000 µM + CEL 1.25 µM	83%	72%
MEL 2000 µM + WOG 25 µM	62%	47%
MEL 2000 µM + WOG 50 µM	78%	78%

**Table 2 ijms-24-09544-t002:** Percentage of apoptotic cells when treated with a combination of drugs. Abbreviations: Iry (irinotecan), Mel (melatonin), Cel (celastrol), Wog (wogonin).

Irinotecan or Melatonin Alone	Percentage of Apoptotic Cells	
	LOVO Cells	LOVO/DX Cells
IRY 5 µM	37%	14%
IRY 20 µM	39%	11%
MEL 2000 µM	37%	16%
Combination of drugs		
IRY 5 µM + WOG 25 µM	49%	20%
IRY 5 µM + MEL 2000 µM	46%	17%
IRY 20 µM + WOG 25 µM	49%	19%
IRY 20 µM + CEL 0.625 µM	47%	26%
IRY 20 µM + CEL 1.25 µM	45%	35%
MEL 2000 µM + WOG 25 µM	50%	23%
MEL 2000 µM + CEL 0.625 µM	46%	19%
MEL 2000 µM + CEL 1.25 µM	39%	30%

**Table 3 ijms-24-09544-t003:** Results of combined drugs in the scratch assay. Abbreviations: Iry (irinotecan), Mel (melatonin), Cel (celastrol), Wog (wogonin).

Irinotecan or Melatonin Alone	Gap Closure	
	LOVO Cells	LOVO/DX Cells
IRY 5 µM	47.6%	23.1%
IRY 20 µM	−2.2% *	3.4%
MEL 2000 µM	57.0%	36.0%
Combination of Drugs		
IRY 5 µM + WOG 25 µM	−0.3% *	19.2%
IRY 5 µM + MEL 2000 µM	48.2%	30.8%
IRY 20 µM + WOG 25 µM	−2.9% *	1.7%
IRY 20 µM + CEL 0.625 µM	0.6%	4.5%
IRY 20 µM + CEL 1.25 µM	−1.0% *	4.7%
MEL 2000 µM + WOG 25 µM	25.0%	7.7%
MEL 2000 µM + CEL 0.625 µM	59.1%	36.3%
MEL 2000 µM + CEL 1.25 µM	56.5%	32.9%

* Results with a negative number showed that the gap increased after 48 h of treatment.

## Data Availability

The data that support the findings of this study are openly available at the request of the reader.

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
