# Peer review of "Combination of Irinotecan and Melatonin with the Natural Compounds Wogonin and Celastrol for Colon Cancer Treatment"

_ijms, 2023, doi:10.3390/ijms24119544_

Round 1

Reviewer 1 Report

The work by Radajewska et al. describe the effect of combinations of Irinotecan or Melatonin with natural compounds including wogonin and celastrol on LoVo cancer cells. They analyze the increment in cytotoxicity, apoptosis and the decrease in cell migration, observing no differences in spheroid growth. The work is well designed and performed and represents an advancement in the knowledge of the field.

There are some minor details that I suggest to modify to improve the presentation and understanding of the work, that I described below:

Minor:

In line 186, it should say in Figure 2, not 1 (Fig. 1 is the representation of the compounds formulae).

The numbering of the concentrations in the x-axis of Fig 2A is not very clear. For instance, Mel2 is 2 mM, and the rest of the concentrations are in µM. Maybe it should be marked as Mel 2000 to be consistent. The same for Fig. 2B.

Table 1 in fact is a repetition of the data presented in Figures 3 and 4. I understand that the goal of the table is to show in a more precise way the differences in mortality of the combinations. In that case, my suggestion is to place Table 1 after Figures 4 and 5, and then include also in this table the values of cytotoxicity produced by IRY and MEL alone to better compare the effect of the combinations.

The same for Table 2 and Table 3. Include the values of apoptosis  or inhibition of cell migration, respectively, produced by IRY and MEL alone.

English in general is fine.

It would be advisable to revise the grammar by an English native speaker

Author Response

Thank you for your comments. We have included our answer in the file below.

Reviewer 2 Report

Comments:

1-      Some English editing is required. The paragraphing of the manuscript is also not acceptable.

2-      To provide a better introduction, try to give readers a general insight into cancer therapy and the role of nanoparticles. For this, you could read and cited the following studies” 1) Increasing the colon cancer cells sensitivity toward radiation therapy via application of Oct4–Sox2 complex decoy oligodeoxynucleotides, 2) Application of decoy oligodeoxynucleotides strategy for inhibition of cell growth and reduction of metastatic properties in nonresistant and Erlotinib‐resistant SW480 cell line, 3) Anticancer evaluation of methotrexate and curcumin-coencapsulated niosomes against colorectal cancer cell lines, 4) Anticancer effect of X-Ray triggered methotrexate conjugated albumin coated bismuth sulfide nanoparticles on SW480 colon cancer cell line.

3-      Provide enough referencing in the material and methods section, especially in the spheroid formation and scratch assays.

4-      In the discussion section, try to discuss combination therapy and other methods.

5-      What software did you use for statistical analysis?

6-      The cat. number of the used products must be added.

7-      Please improve the quality of all Figures (at least resolution on 300 dpi).

8-      What were the reasons for choosing the mentioned concentrations for the drugs used in the MTT test? And why is the MTT test done in 72 hours?

9-      What formula have you used to analyze the scratch test photos? Please mention it in the scratch test method section.

10-  Please analyze the results related to apoptosis so that the amount of primary and secondary apoptosis is reported separately.

Some English editing is required. The paragraphing of the manuscript is also not acceptable.

Author Response

Thank you for your comments. We have included our answers in the file below. 

Reviewer 3 Report

Anna et al., in this manuscript titled "Combination of irinotecan and melatonin with natural com3 pounds – wogonin and celastrol for colon cancer treatment" have described anticancer potential of irinotecan, a standard anticancer chemotherapy agent, in combined therapy with melatonin, wogonin, or celastrol on drug-sensitive colon cancer cells (LOVO cell line) and doxorubicin-resistant colon cancer stem-like cells (LOVO/DX cell sub20 line). The work scientifically sounds great. However, there are wide scope for improvement. Following are the comments need to be addressed before acceptance.

1. Abstract need to drastically improved and more meaningful.

2. Keywords should be restricted to five.

3. Figure 1 should be re-drawn. Maintain uniformity, especially the size of aromatic ring.

4. Sub-sections in the section 2 should be numbered.

5. References can be re-framed by including recent reports.

Minor English editing is necessary. 

Author Response

(The authors gave the same response as above.)
